# Modulational Instability of Ion-Acoustic Waves in Pair-Ion Plasma

**Sharmin Jahan \*, Rubaiya Khondoker Shikha, Abdul Mannan** 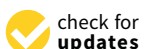 **and A A Mamun** 

Department of Physics, Jahangirnagar University, Savar, Dhaka 1342, Bangladesh;
shikha261phy@gmail.com (R.K.S.); abdulmannan@juniv.edu (A.M.); mamun_phys@juniv.edu (A.A.M.)
\* Correspondence: jahan88phy@gmail.com

**Abstract:** The modulational instability (MI) of ion-acoustic waves (IAWs) is examined theoretically in a four-component plasma system containing inertialess electrons featuring a non-thermal, non-extensive distribution, iso-thermal positrons, and positively as well as negatively charged inertial ions. In this connection, a non-linear Schrödinger equation (NLSE), which dominates the conditions for MI associated with IAWs, is obtained by using the reductive perturbation method. The numerical analysis of the NLSE reveals that the increment in non-thermality leads to a more unstable state, whereas the enhancement in non-extensivity introduces a less unstable state. It also signifies the bright (dark) ion-acoustic (IA) envelope solitons mode in the unstable (stable) domain. The conditions for MI and its growth rate in the unstable regime of the IAWs are vigorously modified by the different plasma parameters (viz., non-thermal, non-extensive $q$-distributed electron, iso-thermal positron, the ion charge state, the mass of the ion and positron, non-thermal parameter $\alpha$, the temperature of electron and positron, etc.). Our findings may supplement and add to prior research in non-thermal, non-extensive electrons and iso-thermal positrons that can co-exist with positive as well as negative inertial ions.

**Keywords:** pair-ion plasma; NLSE; ion-acoustic waves; reductive perturbation method

## 1. Introduction

The physics of pair-ion (PI) plasmas, which have unusual thermodynamic features due to the presence of solely positively and negatively charged species of equal mass [1], has received tremendous attention in recent years as their applications have progressed from the astronomical realm to the terrestrial laboratory. PI plasma can be observed in solar wind [2], $(X_e, F^-)$ [3], positive and negative fullerene ions ($C_{60}^+$ and $C_{60}^-$) [4], $(K^+, SF_6^-)$ [5,6], etc. The formation of PI plasmas (Fullerene ($C_{60}^{\pm}$)) in the laboratory, introduced by Oohara and Hatakeyama [7], has not only proved to be a viable alternative to electron–positron (EP) plasma, but it has also made it more exciting and intriguing to study EP plasma properties in terrestrial situations. The EP plasmas are commonly observed in active galactic nuclei [8], in the early universe [9], and in pulsars [10], etc., and are also being produced in the laboratory [11]. The study of linear and non-linear wave processes in EP plasmas has received a lot of attention in recent years [12]. The PI plasma is expected to be utilized in nanotechnology and for the synthesis of dimers directly from carbon allotropies [13]. Since then, the appeal of PI plasmas is contributing to a great deal of attention from researchers [1,12].

Maxwellian velocity distribution usually describes the thermal equilibrium state of particles, which may not be appropriate for interpreting the dynamics of highly energetic particles. Renyi [14] was the first to suggest the non-extensive $q$ distribution as a way to describe the dynamics of these highly energetic particles, and Tsallis [15] proved the further developments of a $q$ distribution, which are applicable to a broader range in solid-state physics [16], information theory [17], non-equilibrium systems [16], plasma

physics [16], etc. In this distribution, the entropic index $q$ quantifies the extent of non-extensivity. It is worth noting that $q = 1$ refers to Maxwellian behavior, and $q < 1$ ($q > 1$) refers to super-extensivity (sub-extensivity), respectively. It has already been mentioned that energetic electrons exist in a variety of astrophysical plasma domains, with non-Maxwellian components [18]. Non-thermal electrons have been found in the upper ionosphere of Mars [19], in the Earth's bow-shock [20], in the magnetospheres of Jupiter and Saturn [21], and in the vicinity of the Moon [18]. On the basis of observational data from the Freja satellite [22] and the Viking spacecraft [23], Cairns et al. [17] proved that the ubiquitous presence of non-thermal electron distributions in plasma systems can alter the characteristics of IA solitary (IAS) structures. However, to broaden the work of the Cairns non-thermal distribution, Tribeche et al. [24] presented a completely new hybrid (Cairns–Tsallis) distribution inside the theoretical structure of Tsallis non-extensive statistics [15]. The non-thermal, non-extensive electrons can be described by the non-extensive parameter $q$ and the non-thermal parameter $\beta$ (which identifies the degree of non-thermality in plasma species). The primary benefit of employing such a distribution is that it claims to provide increased parametric flexibility in modeling and fitting to a wide range of non-thermal plasmas [24]. Such a kind of hybrid distribution provided an enormous effort to generate various kinds of non-linear phenomena, namely, modulational instability (MI) [16], envelope solitons [25], gigantic waves [16], etc. However, it is worth mentioning that some recent theoretical work spotlighted the effects of non-thermal, non-extensivity on different types of non-linear processes [16,24,26].

The investigation of MI and the associated non-linear structures (e.g., envelope solitons [25], monster waves [16], etc.) has been one of the most popular research topics in recent decades. MI is the carrier wave self-interaction, which is a well-known harmonic generating mechanism, that causes amplitude modulation in non-linear wave propagation [25]. The reductive perturbation method (RPM), used to derive the KdV equation, describes the evolution of non-modulated waves, i.e., a bare pulse with no fast oscillations inside the packet. A well-known non-linear mechanism involved in plasma wave dynamics is amplitude modulation (which may be due to parametric wave coupling), to an interaction between high- and low-frequency modes, or simply to the non-linear self-interaction of the carrier wave. The standard method for studying this mechanism adopts a multiple scales perturbation technique (also known as RPM) [27–29], which generally leads to a non-linear Schrödinger equation (NLSE) describing the evolution of a slowly varying wave packet envelope. Under certain conditions, the wave may undergo a Benjamin–Feir-type MI, i.e., its envelope may collapse under the influence of external perturbations. The MI of wave packets in plasmas acts as a precursor to the formation of bright envelope solitons or highly energetic rogue waves; otherwise, the dark envelope solitons may be formed. In addition, the envelope soliton may be defined as a rapidly oscillating wave that propagates with a characteristic of constant shape and that can be pictured as cut off by a smoothly modulating envelope. Recently, several authors have studied the MI and envelope systems in various types of plasma systems [16,24,26]. Bouzit et al. [16] employed a $q$-non-extensive, non-thermal electron velocity distribution to explore the MI of IAWs, and discovered that plasma supports both the bright and dark envelope solitons. They also discovered that the valid domain for the wave number $k$, at which instabilities occur, differs depending on both the entropic index $q$ and the non-thermal parameter $\alpha$. Bencheriet et al. [30] investigated tiny-amplitude ion-acoustic solitary (IAS) waves in a plasma system containing positive-negative ions and non-thermal electrons, finding that only rarefactive waves are maintained. Tribeche et al. [24] reported IAS waves in a plasma with non-thermal electrons featuring Tsallis distributions, and observed that their plasma model supports the co-existence of smooth rarefactive and spiky compressive IAS waves. As far as we are concerned, there has been no attempt to examine the MI, the related dark and bright envelope solitons, nor the growth rate analysis associated with IAS waves in a four-component PI plasma system. The goal of this study is to enhance Tribeche's [24] work by examining the conditions for the MI in a four-component plasma system, using the

RPM of the IAS waves (in which inertia is provided by the ion masses and restoring force is regulated by the thermal pressure of non-thermal, non-extensive $q$-distributed electrons and iso-thermal positrons).

The manuscript is ordered according to the following scheme: the model equations containing non-thermal, non-extensive electrons, as well as iso-thermal positrons in a PI plasma system, in association with the derivation of the NLSE, are manifested in Section 2. The stability of IAWs and the associated envelope solitons are provided in Sections 3 and 4, respectively. Lastly, the summary of our discussion is delivered in Section 5.

## 2. Model Equations

We have considered an unmagnetized, fully ionized, and four-component PI plasma model consisting of inertial positive ions (charge $q_{+i} = +eZ_{+i}$; mass $m_{+i}$), following the fluid–momentum equations, inertial negative ions (charge $q_{-i} = -eZ_{-i}$; mass $m_{-i}$), explained by the fluid–dynamic equations, inertialess electrons (charge $q_e = -e$; mass $m_e$), assumed to obey a non-thermal, non-extensive distribution, and an inertialess positron, denoted by $n_p$ (charge $q_p = +e$; mass $m_p$), that follows an iso-thermal distribution. The quasi-neutrality criterion is maintained in our model, which can be stated as $n_{e0} + Z_{-i}n_{-i0} = n_{p0} + Z_{+i}n_{+i0}$. Now, the fundamental set of normalized equations can be depicted as follows:

$$\frac{\partial n_{+i}}{\partial t} + \frac{\partial}{\partial x}(n_{+i}u_{+i}) = 0, \tag{1}$$

$$\frac{\partial u_{+i}}{\partial t} + u_{+i}\frac{\partial u_{+i}}{\partial x} + \frac{\partial \phi}{\partial x} = 0, \tag{2}$$

$$\frac{\partial n_{-i}}{\partial t} + \frac{\partial}{\partial x}(n_{-i}u_{-i}) = 0, \tag{3}$$

$$\frac{\partial u_{-i}}{\partial t} + u_{-i}\frac{\partial u_{-i}}{\partial x} - \gamma_1\frac{\partial \phi}{\partial x} = 0, \tag{4}$$

$$\frac{\partial^2 \phi}{\partial x^2} = n_e(1 + \gamma_2 - \gamma_3) + \gamma_3 n_{-i} - \gamma_2 n_p - n_{+i}, \tag{5}$$

The following are the normalization and related parameters: $n_{+i} = N_{+i}/n_{+i0}$, $n_{-i} = N_{-i}/n_{-i0}$, $n_e = N_e/n_{e0}$, $n_p = N_p/n_{p0}$, $u_{+i} = U_{+i}/C_{+iD}$, $u_{-i} = U_{-i}/C_{+iD}$, $x = X/\lambda_{+iD}$, $t = T\omega_{+ip}$, $\phi = e\varphi/k_B T_e$, $C_{+iD} = (Z_{+i}k_B T_e/m_{+i})^{1/2}$, $\omega_{+ip} = (4\pi e^2 Z_{+i}^2 n_{+i0}/m_{+i})^{1/2}$, $\lambda_{+iD} = (k_B T_e/4\pi e^2 Z_{+i}n_{+i0})^{1/2}$, $\gamma_1 = Z_{-i}m_{+i}/Z_{+i}m_{-i}$, $\gamma_2 = n_{p0}/Z_{+i}n_{+i0}$, and $\gamma_3 = Z_{-i}n_{-i0}/Z_{+i}n_{+i0}$, where $n_{+i}$, $n_{-i}$, $n_e$, and $n_p$ stand for the number densities of the positive ions, negative ions, electrons, and positrons, respectively. Conversely, $u_{+i}$, $u_{-i}$, $x$, $t$, $\phi$, $C_{+iD}$, $\omega_{+ip}$, $\lambda_{+iD}$, $k_B$, and $T_e$ define the positive ion fluid speed, negative ion fluid speed, space co-ordinate, time co-ordinate, electro-static potential, sound speed of the positive ions, angular frequency of the positive ions, Debye length of the positive ions, Boltzmann constant, and electron temperature, respectively. The number densities of the electrons (obeying a non-thermal, non-extensive distribution [24]) and the positrons (following an iso-thermal distribution [31]) can now be expressed using the normalized equations below:

$$n_e = (1 + A\phi + B\phi^2)[1 + (q-1)\phi]^{\frac{q+1}{2(q-1)}}, \tag{6}$$

$$n_p = \exp(-\gamma_4 \phi). \tag{7}$$

where $A = -16q\alpha/(3 - 14q + 15q^2 + 12\alpha)$ (with $q$ ($\alpha$) as the non-extensive (non-thermal) parameter, respectively. $\alpha$, the non-thermal parameter that determines the proportion of the fast energetic particles and thermal electrons [32], $B = A(1 - 2q)$, and $\lambda_4 = T_e/T_p$ (with $T_p$

($T_e$) being the temperature of the positron (electron), respectively and $T_e > T_p$)). Now, by substituting (6) and (7) into (5) and expanding up to the third order, we can write:

$$\frac{\partial^2 \phi}{\partial x^2} + n_{+i} + \gamma_3$$
$$= 1 + \gamma_3 n_{-i} + \frac{[(1 + \gamma_2 - \gamma_3)\Pi_1 + 2\gamma_2\gamma_4]}{2}\phi$$
$$+ \frac{[(1 + \gamma_2 - \gamma_3)\Pi_2 - 4\gamma_2\gamma_4^2]}{8}\phi^2$$
$$+ \frac{[(1 + \gamma_2 - \gamma_3)\Pi_3 + 8\gamma_2\gamma_4^3]}{M_3}\phi^3 + \cdots \tag{8}$$

where $\Pi_1 = 2A + q + 1$, $\Pi_2 = 8B + 4Aq + 4A + 2q - q^2 + 3$, and $\Pi_3 = 24Bq + 24B + 12q - 6q^2 + 18 + 3Q^3 + 3q^2 - 9q^2 - 9q - 5q^2 - 5q - 15q - 15$. In order to study the stable and unstable domains of the IAWs' PI plasma, we introduce the following stretched coordinates, according to the reductive perturbation method (multiscale technique): [27–29,32,33]:

$$\xi = \epsilon(x - V_g t), \tag{9}$$
$$\tau = \epsilon^2 t, \tag{10}$$

where $V_g$ denotes the group speed and $\epsilon$ denotes a minor parameter. The dependent variables [33] can, therefore, be stated as follows:

$$Y(x,t) = Y_0 + \sum_{m=1}^{\infty} \epsilon^{(m)} \sum_{l=-\infty}^{\infty} Y_{il}^{(m)}(\xi,\tau)e^{il(kx-\omega t)}, \tag{11}$$

where $Y_{il}^{(m)} = [n_{+il}^{(m)}, u_{+il}^{(m)}, n_{-il}^{(m)}, u_{-il}^{(m)}, \phi_l^{(m)}]^T$, $Y_0 = [1, 0, 1, 0, 0]^T$, and $k$ ($\omega$) is a real variable that represents the number of carrier waves (frequency), respectively. The dispersion relation can be obtained by substituting these expansions into the motion equations:

$$\omega^2 = \frac{k^2(1 + \gamma_1\gamma_3)}{(k^2 + M_1)}. \tag{12}$$

We have numerically analyzed Equation (12) in order to explain the linear dispersion features of IAWs for different values of $\gamma_1$. The outcomes are depicted in Figure 1, which manifest that (a) for the lower range of $k$, the IAW mode grows exponentially with it, but saturation sets in after a specific value of $k$, and that (b) the wave frequency ($\omega$) rises exponentially with the positive ion mass for a fixed value of other plasma parameters. The group velocity,

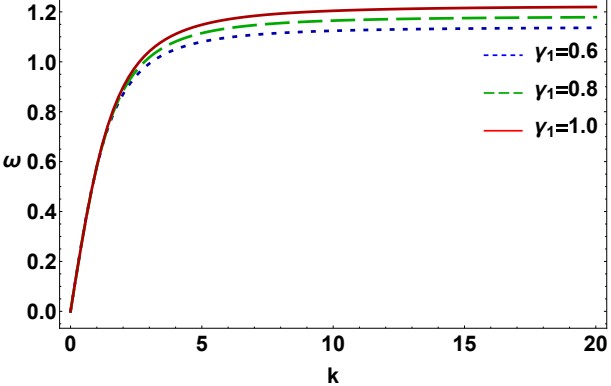

**Figure 1.** The variation of $\omega$ with $k$ for several values of $\gamma_1$, along with $\alpha = 0.5$, $\gamma_2 = 1.2$, $\gamma_3 = 0.5$, $\gamma_4 = 1.4$, and $q = 1.5$.

$$V_g = \frac{\omega(1 + \gamma_1\gamma_3 - \omega^2)}{k(1 + \gamma_1\gamma_3)}, \tag{13}$$

and, finally, the NLSE,

$$i\frac{\partial\Phi}{\partial\tau} + P\frac{\partial^2\Phi}{\partial\xi^2} + Q\Phi|\Phi|^2 = 0, \tag{14}$$

where $\Phi = \phi_1^{(1)}$ for simplicity, which denotes the electric potential correction [34] $P = (3V_g^2/2\omega k) - (3V_g/2k)$, $Q = M_{14}/(2k^2 + 2k^2\gamma_1\gamma_3)$, and the other parameters are:

$$\begin{aligned}
M_{14} = {}& 3M_3\omega^3 + 2M_2\omega^3 M_8 + 2M_2\omega^3 M_{13} - \omega k^2 M_4 \\
& - \omega k^2 M_9 - 2k^3 M_5 - 2k^3 M_{10} - \gamma_1\gamma_3\omega k^2 M_6 \\
& - \gamma_1\gamma_3\omega k^2 M_{11} - 2\gamma_1\gamma_3 k^3(M_7 + M_{12}).
\end{aligned}$$

$$\begin{aligned}
M_4 = {}& (3k^4/2\omega^4) + (2M_8\omega^2 k^2/2\omega^4), \\
M_5 = {}& (k^3/2\omega^3) + (M_8 k/\omega), \\
M_6 = {}& (3\gamma_1^2 k^4/2\omega^4) - (\gamma_1 M_8 k^2/\omega^2), \\
M_7 = {}& (\gamma_1^2 k^3/2\omega^3) - (\gamma_1 M_8 k/\omega), \\
M_8 = {}& (M_2\omega^2/k^2 + \gamma_1\gamma_3 k^2 - M_1\omega^2 - 4k^2\omega^2) + (3\gamma_3\gamma_1^2 k^4 \\
& - 3k^4/2\omega^2 k^2 + 2\gamma_1\gamma_3\omega^2 k^2 - 2M_1\omega^4 - 8k^2\omega^4) \\
M_9 = {}& (2k^3/V_g\omega^3) + (k^2/V_g^2\omega^2) + (M_{13}/V_g^2), \\
M_{10} = {}& (k^2/V_g\omega^2) + (M_{13}/V_g), \\
M_{11} = {}& (2\gamma_1^2 k^3/V_g\omega^3) + (\gamma_1^2 k^2/V_g^2\omega^2) - (\gamma_1 M_{13}/V_g^2), \\
M_{12} = {}& (\gamma_1^2 k^2/V_g\omega^2) - (\gamma_1 M_{13}/V_g), \\
M_{13} = {}& [(2M_2 V_g^2\omega^3 + 2\gamma_3 V_g\gamma_1^2 k^3 - 2V_g k^3 \\
& + \omega k^2)/(\omega^3 + \gamma_1\gamma_3\omega^3 - M_1 V_g^2\omega^3)] \\
& + [(\gamma_3\gamma_1^2 k^2)/(\omega^2 + \gamma_1\gamma_3\omega^2 - M_1\omega^2 V_g^2)].
\end{aligned}$$

## 3. Stability of IAWs

To study the MI of IAWs, we consider the linear solution of Equation (12) in the form $\Phi = \widetilde{\Phi}e^{iQ|\widetilde{\Phi}|^2\tau} + c.c.$, where $\widetilde{\Phi} = \widetilde{\Phi}_0 + \epsilon\widetilde{\Phi}_1$ and $\widetilde{\Phi}_1 = \widetilde{\Phi}_{1,0}e^{i(\widetilde{k}\xi - \widetilde{\omega}\tau)} + c.c.$ We note that the amplitude depends on the frequency, and that the perturbed wave number $\widetilde{k}$ and frequency $\widetilde{\omega}$ are different from $k$ and $\omega$. Now, substituting these into Equation (12), one can easily obtain the following non-linear dispersion relation [31,34,35]:

$$\widetilde{\omega}^2 = P^2\widetilde{k}^2\left(\widetilde{k}^2 - \frac{2|\widetilde{\Phi}_0|^2}{P/Q}\right). \tag{15}$$

The MI of IAWs, in which the NLSE dominates the amplitude progression, is entirely dependent on two terms: the non-linear ($P$) and dispersive coefficients ($Q$) (see Equation (15)). Both of the coefficients are the functions of different physical plasma parameters, such as $\gamma_1$, $\gamma_2$, $\gamma_3$, $\alpha$, $q$, etc. Outlining $P/Q$ against the wave number ($k$) for different plasma properties can be used to acknowledge the stability conditions of IAWs [36]. The sign of $P/Q$ plays a significant role to recognize the criteria of the IAWs. It is important to mention that IAWs are modulationally stable when the non-linear and dispersive coefficients have different signs ($P/Q < 0$), whereas the instability condition is obtained when $P$ and $Q$ have the same sign ($P/Q > 0$) (see Equation (15)) [36]. It is necessary to note that the critical

threshold number, ($k_c$), is defined as the intersecting point at which a stable and unstable domain can be obtained for IAWs. Additively, the term plays a decisive role in order to differentiate between stable and unstable regions of IAWs [36].

The impact of $\gamma_2$ on $k_c$ in the variation of the $P/Q$ curve with $k$ is displayed in Figures 2 and 3, respectively. It is obvious from the figures that: (a) both the modulationally stable and unstable domains can be observed in non-linear and dispersive IAWs (Figures 2 and 3); (b) the IAWs are modulationally unstable for a trivial value of $k$ ($k = k_c \cong 0.4$) when the other parameters remain constant (Figure 2). On the other hand, in Figure 3, the instability condition can be observed at a value of $k = k_c = 0.5$; (c) the stable region of the IAWs increases (decreases) with the equilibrium number density of the positron (positive ion) for a constant value of the charge state of the positive ion (via $\gamma_2$) (Figures 2 and 3, respectively); (d) the equilibrium positron number density plays a more major role in enhancing the stability domain of IAWs in the non-Maxwellian case ($q = 1.5, \alpha = 0.5$, as shown in Figure 2) than in the Maxwellian case ($q = 1, \alpha = 0$, as clearly seen in Figure 3). Thus, an excess number of positrons in our considered plasma system leads to the maximization of the stable domain of the wave profile.

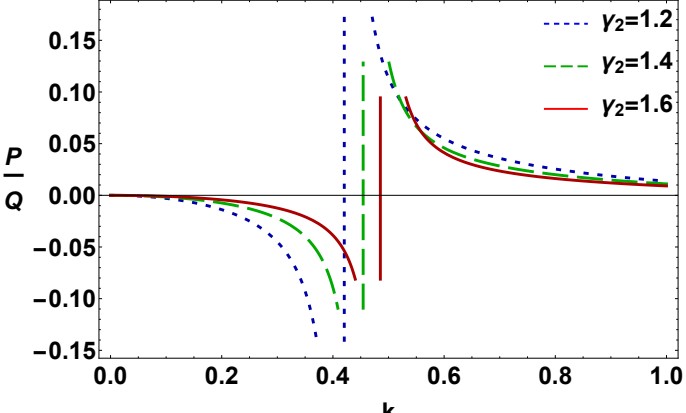

**Figure 2.** The relationship between $P/Q$ and $k$ for various values of $\gamma_2$ (when $\alpha = 0.5$ and $q = 1.5$), along with $\gamma_3 = 0.5$ and $\gamma_4 = 1.4$.

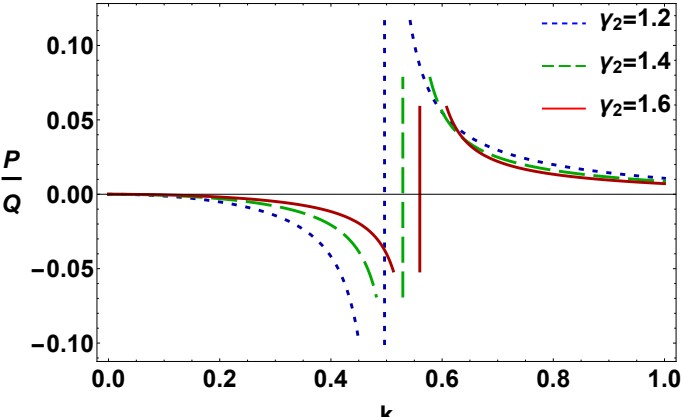

**Figure 3.** For different values of $\gamma_2$, the change of $P/Q$ with $k$ (when $\alpha = 0$ and $q = 1.0$), along with $\gamma_3 = 0.5$ and $\gamma_4 = 1.4$.

The effect of the non-thermal parameter ($\alpha$) and the non-extensive index ($q$) on the stability of the wave profile can easily be recognized from Figures 4 and 5, respectively. The outcomes are as follows: (a) the stable and unstable regions of the IAWs can be noticed in Figures 4 and 5, respectively; (b) it is clear from Figure 4 that the instability domain of the IAWs, which arises by varying $\alpha$ (while keeping other parameters constant), strikes at a value of $k_c = 0.38$, whereas in Figure 5, the same condition commences at a value of

$k_c = 0.4$; (c) the stability of the IAWs declines (rises) by increasing the value of the non-thermal parameter (non-extensive $q$ index) (Figures 4 and 5, respectively), and this result is in good agreement with the work of Ghosh and Banerjee [25]; (d) in order to enhance the stability condition of the IAWs, the non-thermal parameter ($\alpha$) plays a completely opposite character to the non-extensive $q$ index (Figures 4 and 5).

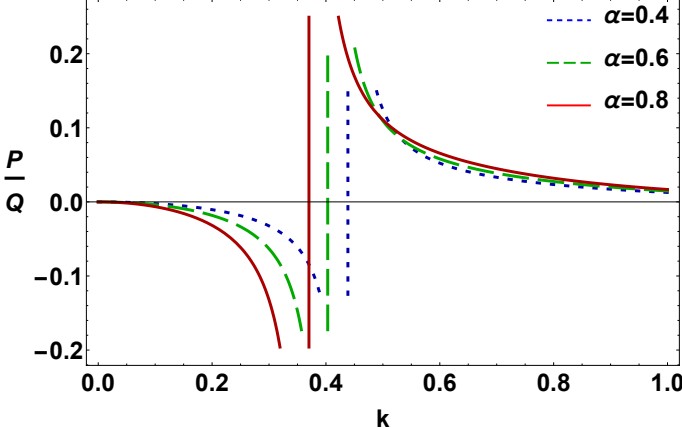

**Figure 4.** Plot of $P/Q$ with $k$ for different values of $\alpha$ (when $q = 1.5$), along with $\gamma_1 = 0.8$, $\gamma_2 = 1.2$, $\gamma_3 = 0.5$, and $\gamma_4 = 1.4$.

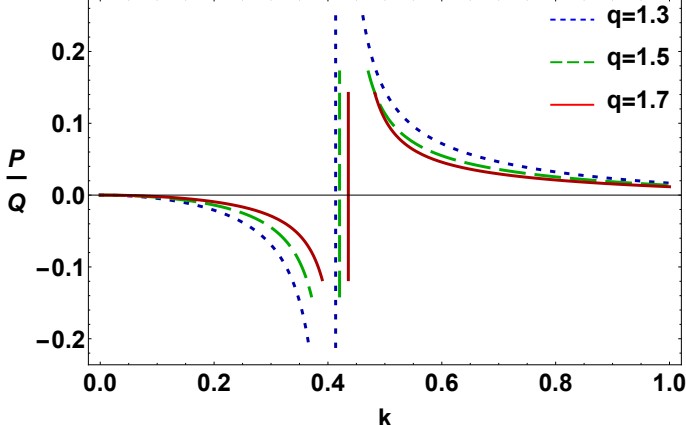

**Figure 5.** The variation of $P/Q$ with $k$ for various values of $q$ (when $\alpha = 0.5$), together with $\gamma_1 = 0.8$, $\gamma_2 = 1.2$, $\gamma_3 = 0.5$, and $\gamma_4 = 1.4$.

The disparity of the growth rate ($\Gamma$) of the MI of IAWs varies with the wave number ($k$) of the changing value of $\gamma_1$, and $\gamma_4$ is displayed in Figures 6 and 7, respectively. However, the graphical representations reveal that (a) the growth rate $\Gamma$ reduces (advances) in the values of $Z_{+i}$ ($Z_{-i}$) (via $\gamma_1$) by keeping the other parameters constant (Figure 6), and that (b) if the value of $T_e$ is increased for a fixed value of $T_p$, then $\Gamma$ decreases (via $\gamma_4$) (Figure 7). Therefore, the non-linearity of our considered plasma is very sensitive to the changes of two parameters, namely, $\gamma_1$ and $T_p$, which cause the maximum growth rate of the wave profile.

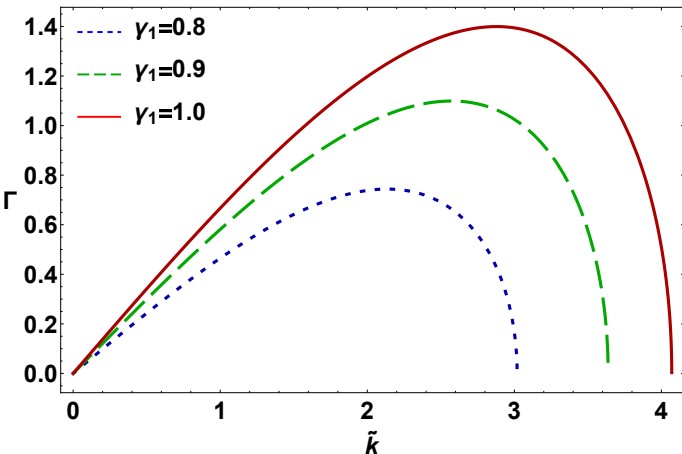

**Figure 6.** Graphical representation of $\Gamma$ with $\tilde{k}$ for different values of $\gamma_1$, along with $\alpha = 0.05$, $\gamma_2 = 1.2$, $\gamma_3 = 0.5$, $\gamma_4 = 1.4$, $q = 1.5$, $\omega$, $\Phi_0 = 0.5$, and $k = 0.6$.

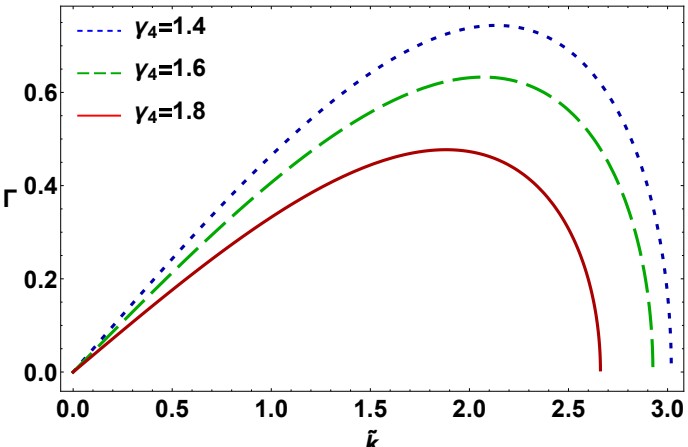

**Figure 7.** Numerical analysis of $\Gamma$ with $\tilde{k}$ for different values of $\gamma_4$, along with $\alpha = 0.05$, $\gamma_1 = 0.8$, $\gamma_2 = 1.2$, $\gamma_3 = 0.5$, $q = 1.5$, $\omega$, $\Phi_0 = 0.5$, and $k = 0.6$.

## 4. Envelope Solitons

The sign of the coefficients ($P$ and $Q$) declares that two types of envelope modes may exist, e.g., bright and dark envelope solitons. The bright envelope solitons have attractive non-linearity [34] with a bell-shaped structure. Envelope solitons in the form of the bright type are found in space plasmas [34]. This type of soliton exists when $P$ and $Q$ have the same sign ($P/Q > 0$), which occurs at larger wave numbers (shorter wavelengths). The localized envelope pulses of the form is shown in Figure 8. The common analytical form of bright envelope modes can be read as [32,34,35,37]:

$$\Phi(\xi, \tau) = \Pi_5 \times \Pi_6, \tag{16}$$

where

$$\Pi_5 = \left[ \psi_0 \, \text{sech}^2 \left( \frac{\xi - U\tau}{W} \right) \right]^{1/2},$$

$$\text{and} \quad \Pi_6 = \exp \left[ \frac{i}{2P} \left\{ U\xi + \left( \Omega_0 - \frac{U^2}{2} \right) \tau \right\} \right],$$

It is noted that, in the $\Pi_5$ term, $\psi_0$ indicates the envelope amplitude, $U$ is the propagation speed of the localized pulse, $W$ is the pulse width, which can be written as $W = \sqrt{(2|P/Q|)/\psi_0)}$, and $\Omega_0$ is the oscillating frequency for $U = 0$.

On the other hand, the dark solitons are dips or holes in a wave background that require repulsive or defocusing non-linearity [30]. In this soliton, *P* and *Q* have the opposite sign $P/Q < 0$ for large wavelengths (or small wave numbers in the modulationally stable region). The localized envelope pulse of the dark envelope soliton is depicted in Figure 9.

The general analytical form for the dark-type mode can be written as [32,34]:

$$\Phi(\xi, \tau) = \Pi_7 \times \Pi_8, \tag{17}$$

where

$$\Pi_7 = \left[\psi_0 \tanh^2\left(\frac{\xi - U\tau}{W}\right)\right]^{1/2},$$

$$\text{and} \quad \Pi_8 = \exp\left[\frac{i}{2P}\left\{U\xi - \left(\frac{U^2}{2} - 2PQ\psi_0\right)\tau\right\}\right].$$

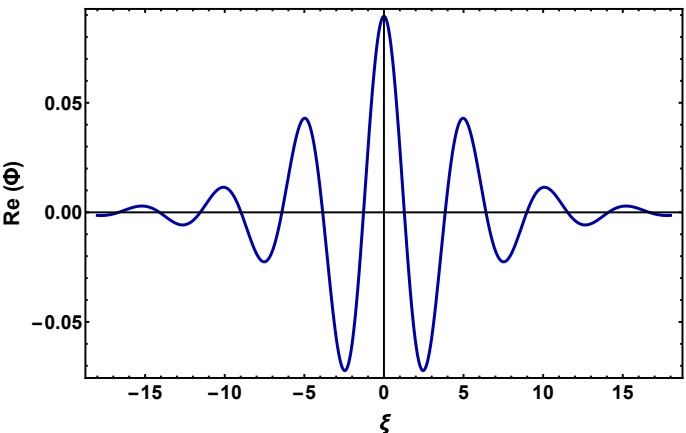

**Figure 8.** The variation of $Re(\Phi)$ with $\xi$, along with $\alpha = 0.5$, $\gamma_1 = 0.8$, $\gamma_2 = 1.2$, $\gamma_3 = 0.5$, $\gamma_4 = 1.4$, and $q = 1.5$, $\Phi_0 = 0.008$, $\psi_0 = 0.008$, $\tau = 0$, $k = 0.6$, and $U = 0.4$.

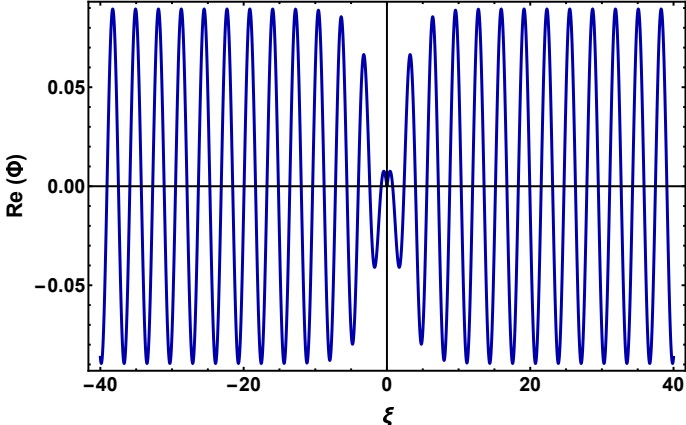

**Figure 9.** The variation of $Re(\Phi)$ with $\xi$, along with $\alpha = 0.5$, $\gamma_1 = 0.8$, $\gamma_2 = 1.2$, $\gamma_3 = 0.5$, $\gamma_4 = 1.4$, and $q = 1.5$, $\Phi_0 = 0.008$, $\psi_0 = 0.008$, $\tau = 0$, $k = 0.3$, and $U = 0.4$.

## 5. Conclusions

We have examined the basic features of IAWs in an unmagnetized PI plasma system containing a non-thermal, non-extensive *q*-distributed electron, an iso-thermal positron, and positively as well as negatively charged inertial ions. A multiscale technique (reductive perturbation method) is employed to deduce the NLSE. From the investigation, it can be seen that both the modulationally stable and unstable domains can be observed in

non-linear and dispersive IAWs. The wave frequency ($\omega$) grows exponentially with the rising value of the positive ion mass. It is worth mentioning that the equilibrium positron number density plays a more vital role in enhancing the stability domain of IAWs in the non-Maxwellian case than in the Maxwellian case. Moreover, in order to enhance the stability condition of the IAWs, the non-thermal parameter ($\alpha$) plays a completely opposite character to the non-extensive $q$ index. The increment of both the negative ion mass and the electron temperature in the PI plasma system tend to decrease the growth rate of IAWs. Note that the findings of our present investigation will be useful for understanding the non-linear phenomena (viz., the MI of IAWs and the formation of envelope modes) in IAWs, where the electrons follow the non-thermal, non-extensive distribution, and the positrons obey the iso-thermal distribution.

**Author Contributions:** Author Contributions: Methodology, S.J., A.M.; Calculation, S.J.; Graphical analysis, S.J., R.K.S.; Paper writing, S.J., R.K.S.; Supervision, A.A.M. All authors have read and agreed to the published version of the manuscript.

**Funding:** This research received no external funding.

**Acknowledgments:** Sharmin Jahan gratefully acknowledges the NST (National Science and Technology) Fellowship for their financial support to the completion of this work.

**Conflicts of Interest:** The authors declare no conflict of interest.

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
