# Peer review of "Modulational Instability of Ion-Acoustic Waves in Pair-Ion Plasma"

_plasma, doi:10.3390/plasma5010001_

Round 1

Reviewer 1 Report

In this work modulational instability of ion-acoustic waves is examined theoretically in a four-component plasma system: electrons, non-thermal non-extensive electrons, isothermal positrons and inertial positively and negatively charged ions. The work was done at a high scientific level and the paper is recommended to be published in the Plasma.

Author Response

Response to Reviewer 1

Point 1: In this work modulational instability of ion-acoustic waves is examined theoretically in a four-component plasma system: electrons, non-thermal non-extensive electrons, isothermal positrons and inertial positively and negatively charged ions. The work was done at a high scientific level and the paper is recommended to be published in the Plasma.

Response 1: We thank very much the reviewer for accepting our manuscript.

Reviewer 2 Report

The topic is interesting but it is necessary to clarify some points

abstract : explain what does it mean modulation instability

line 94 there are formulas outside the text

lines 102-103 q(\alpha) extensive (non-thermal parameter) must be explained: what does mean extensive non-thermal? Why there is an \alpha in the formula for A. Explain the choice and meaning of A and B

130 In what sense do the figures (2) and (3) show modulation stability and instability?

line 123 the symbol \ph^1_1 is not defined

line 170 what are bright and dark envelope solitons? Explain the terminology and the concept

line 188 you mention multiscale technique but I do not see where they are

195 why \alpha is considered a non-thermal parameter?

199 explain in more detail which non-linear phenomena  will be explained with your result

200 extensive is written wrong

Author Response

Response to Reviewer 2

Point 1:The topic is interesting but it is necessary to clarify some points. Abstract:explain what does it mean modulation instability?

Response1: The reductive perturbation method used to derive the KdV equation, describes the evolution of a non-modulated waves, i.e., a bare pulse with no fast oscillations inside the packet. A well known nonlinear mechanism involved in plasma wave dynamics is amplitude modulation, which may be due to parametric wave coupling, interaction between high and low frequency modes or simply to the nonlinear self-interaction of the carrier wave. The standard method to study this mechanism adopts a multiple scales perturbation technique (also known as reductive perturbation method)[27-29],
which generally leads to a nonlinear Schr\"odinger equation (NLSE) describing the evolution of a slowly varying wave wavepacket envelope. Under certain conditions, the wave may undergo a Benjamin-Feir-type modulational instability
(MI), i.e., its envelope may collapse under the influence of external perturbations. The MI of wave packets in plasmas acts as a precursor for the formation of bright envelope solitons or highly energetic rogue waves, otherwise the dark envelope solitons may be formed.

Point 2: Line 94 there are formulas outside the text.

Response 2: We have fixed this technical or typo error in our revised manuscript.

Point 3: Lines 102$-$103 $q(\alpha)$ extensive (non-thermal parameter) must be explained: what does mean extensive non-thermal? Why there is an $\alpha$ in the formula for A. Explain the choice and meaning of A and B.

Response 3: Maxwellian velocity distribution usually describes the thermally equilibrium state of particles, which may not be appropriate for interpreting the dynamics of highly energetic particles. Renyi [14] was the first to suggest the
non-extensive $q$-distribution as a way to describe the dynamics of these highly energetic particles, and Tsallis [22] proved further development of $q$-distribution. It is worth noting that $q = 1$ refers to Maxwellian behavior, and $q < 1$ $(q > 1)$ refers to super-extensivity (sub-extensivity). Cairns
et al. [20] first proposed a distribution that uses a parameter $\alpha$ to illustrate the concept of high-energy tails, explaining how to measure the deviation from the isothermal or Maxwellian distribution function. (Please see paragraph 2 in introduction section). Tribeche \textit{et al.} [24] reported the ion-acoustic waves in a plasma with non-thermal electrons featuring Tsallis distribution and observed that their plasma model supports the co-existence of smooth rarefactive and spiky compressive ion-acoustic waves. This led them to believe that non-thermality and non-extensivity may have a simultaneous effect on the nature (rarefactive or compressive) of ion-acoustic structures.\\ Please note that there is no physical reason for defining the terms $A$ and $B$. We have defined $A$ and $B$ terms to keep equation (6) in a single line. 

Point 4: 130 In what sense do the figures (2) and (3) show modulation stability and instability?

Response 4: To answer the question, we have highlighted a paragraph in the section of stability of IAWs. (Kindly see the first paragraph in Sec.3 in the stability of IAWs).

Point 5: Line 123 the symbol $\phi^1_1$ is not defined

Response 5: We have written $\Phi = \phi_1^{(1)}$ for simplicity. However, $\phi$ is a electrostatic potential of ion-acoustic wave and it is defined after equation (5).

Point 6: Line 170 what are bright and dark envelope solitons? Explain the terminology and the concept

Response 6: Depending on the sign of the coefficients $P$ and $Q$, two types of envelope solitonic solutions, which gives rise to bright and dark envelope solitons, can be obtained [32,34,35,36]. The stable bright envelope solitons are formed in $PQ >0$ region due to the balance between the nonlinearity and the dispersion. On the other hand, in the same region ($PQ >0$), the unstable bright envelope solitons are formed when the nonlinearity is so high that the
dispersion can not balance it. The condition for the dark envelope solitons is $PQ <0$.

Point 7: Line 188 you mention multiscale technique but I do not see where they are

Response 7: The reviewer is right. It was our mistake. We have mentioned this point before equation (9).

Point 8: 195 why $\alpha$ is considered a non-thermal parameter?

Response 8: $\alpha$ is the non-thermal parameter that determines the proportion of the fast energetic particles and thermal electrons [32].

Point 9: 199 explain in more detail which nonlinear phenomena will be explained with your result?

Response 9: We have incorporated this point in our revised manuscript.

Point 10: 200 extensive is written wrong

Response 10: We have corrected this typo-error in our revised manuscript.
